# Trade-Offs among Immune Mechanisms: Bacterial-Challenged *Spodoptera frugiperda* Larvae Reduce Nodulation Reactions during Behavioral Fever

**DOI:** 10.3390/insects14110864

**Published:** 2023-11-09

**Authors:** Lei Zhang, Cynthia L. Goodman, Joseph A. Ringbauer, Xingfu Jiang, Weixiang Lv, Dianjie Xie, Tamra Reall, David Stanley

**Affiliations:** 1State Key Laboratory for Biology of Plant Diseases and Insect Pests, Institute of Plant Protection, Chinese Academy of Agricultural Sciences, Beijing 100193, China; leizhang@ippcaas.cn (L.Z.); lvwx@cwnu.edu.cn (W.L.); xiedianjie2023@163.com (D.X.); 2Biological Control of Insects Research Laboratory, USDA, Agricultural Research Service, Columbia, MO 65203, USA; goodmanc@missouri.edu (C.L.G.); joseph.ringbauer@usda.gov (J.A.R.J.); 3Key Laboratory of Southwest China Wildlife Resources Conservation, China West Normal University, Nanchong 637002, China; 4MU Extension, University of Missouri, Kansas City, MO 64014, USA; reallt@missouri.edu

**Keywords:** *Spodoptera frugiperda*, insect immunity, prostaglandins, behavioral fever, nodulation

## Abstract

**Simple Summary:**

Insect innate immunity is composed of cellular and humoral reactions to infections and invasions. Cellular immunity is the first response, launched immediately when an infection is detected and it involves direct interactions between hemocytes and infecting microbes. Some insect species respond to infection with behavioral fever by moving to warmer sites or increasing exposure to heat sources. We predicted that behavioral fevers, which entail changing postures with respect to the sun or moving to warmer locations, but not increasing metabolic rates as in mammalian fevers, would be a relatively low-cost immune function compared to the costs of replacing the many hemocytes lost in immune reactions. Based on this reasoning, we posed the hypothesis that hemocyte-based immunity is traded off for behavioral fevers in infected larvae of the fall armyworm, *Spodoptera frugiperda*, when they can fever. Here, we report that infected larvae that were allowed to fever produced far fewer nodules compared to controls that were not allowed to fever.

**Abstract:**

Insect innate immunity is composed of cellular and humoral reactions, the former acting via circulating hemocytes and the latter via immune signaling that lead to the production of antimicrobial peptides and phenol oxidase-driven melanization. Cellular immunity involves direct interactions between circulating hemocytes and invaders; it includes internalization and killing microbes (phagocytosis) and formation of bacterial-laden microaggregates which coalesce into nodules that are melanized and attached to body walls or organs. Nodulation can entail investing millions of hemocytes which must be replaced. We hypothesized that biologically costly hemocyte-based immunity is traded off for behavioral fevers in infected larvae of fall armyworms, *Spodoptera frugiperda*, that were allowed to fever. We tested our hypothesis by infecting larvae with the Gram-negative bacterium, *Serratia marcescens*, placing them in thermal gradients (TGs) and recording their selected body temperatures. While control larvae selected about 30 °C, the experimental larvae selected up 41 °C. We found that 4 h fevers, but not 2, 6 or 24 h fevers, led to increased larval survival. Co-injections of *S. marcescens* with the prostaglandin (PG) biosynthesis inhibitor indomethacin (INDO) blocked the fevers, which was reversed after co-injections of SM+INDO+Arachidonic acid, a precursor to PG biosynthesis, confirming that PGs mediate fever reactions. These and other experimental outcomes support our hypothesis that costly hemocyte-based immunity is traded off for behavioral fevers in infected larvae under appropriate conditions.

## 1. Introduction

Insect innate immunity is generally assorted into humoral and hemocytic mechanisms; the former is registered as production of anti-microbial peptides (AMPs) and proteolytic activation of prophenoloxidase (PPO) into its active counterpart, phenol oxidase (PO) [1,2]. Newly expressed AMPs appear in the hemolymph of infected insects approximately 6–12 h post-infection (PI). Hemocytic immunity involves direct interactions between circulating hemocytes and invaders. Cellular defense functions include phagocytosis, encapsulation of parasitoids and parasites, and formation of hemocyte nodules. Nodulation clears most infecting microbes from hemolymph circulation within the first 2 h PI [3], during which very large numbers of circulating and sessile hemocytes are lost [4]. These many lost hemocytes may represent substantial biological costs of immune protection, some of which are recorded as fitness reductions. De Roode and Lefere [5] reviewed aspects of a third category of insect immunity, behavioral immunity, such as avoiding infectious situations, grooming, and selecting uninfected mating partners. Such behaviors have been recorded in insects and mammals. 

Aside from humoral and hemocytic immunity, many infected animals protect themselves from sepsis by generating fevers. Fevers help animals, and some plants, deal with invaders by elevating their body temperatures above the invaders’ permissive thresholds [6]. The fevers can kill infecting microbes or slow their development to the extent that cellular and humoral immune functions clear the remaining invaders. There are several fever mechanisms in mammals, including inflammation-associated fever and drug-induced fever, which can harm hosts at the cellular, tissue and systemic levels. Most human fevers are due to sepsis, and they likely provide benefit [7]. Endothermic animals generate fevers by increasing their metabolic rates, often via futile biochemical cycles. Although some insects, such as bumble bees, generate high thoracic temperatures via flight muscle metabolism during flight [6], most insects are ectothermic and they generate fevers through various behaviors, such as moving to warmer zones within habitats or assuming basking positions that increase exposure to heat sources. De Roode and Lefere [5] regard behavioral fever as a subset of behavioral immune mechanisms.

Some insect species reduce pathogen loads by behavioral fever, which has been recorded in various species since the 1940s and 1950s [6]. The fever reactions can enhance fitness by leading to improved recovery in natural settings. Under some circumstances, they can thwart microbe-based biological pest control programs, as seen in the locust *Locusta migratoria* [8]. The positive examples of fever-mediated survival tend to obscure subtle features of behavioral fever [9]. Fever per se can be detrimental, leading to reduced survival and reproductive potentials. Also, lethal temperatures of some invaders can be higher than host fever capacity. While fevers convey fitness benefits in some insect species, it is not a universal facet of insect immunity. Stahlschmidt and Adamo [10] reported that 14 of the 18 insect species they included in their meta-analysis expressed fever to various extents, from about 0.5 to 6 °C. They found that cockroaches did not fever, while crickets and grasshoppers (including locusts) did under some challenges. 

Honey bees fever, but not most fly species, although there may be considerable variations in fever expression among species within orders. Watson et al. [11] recorded behavioral fever reactions in house flies, *Musca domestica*, after fungal, *Entomophthora muscae*, infection. Also, Kalsbeek et al. [12] found more flies infected with the fungal pathogen, *E. muscae* on warm surfaces compared to uninfected controls. They also reported the outcomes of a mark-recapture experiment in which experimental flies were infected with a related pathogen, *E. schizophorae*, and then marked on their dorsal surfaces with a water-based acrylic yellow. The yellow-marked experimental flies were released with blue-marked controls. Compared to controls, they recovered significantly more yellow-marked flies on heat lamps. The authors interpreted these results to show the infected flies gave themselves behavioral fevers by selecting the warmer heat lamps rather than cooler surfaces. The house fly fevers are more subtle than simple on/off responses. Flies infected with another fungal species, *Beauveria bassiana*, also expressed behavioral fevers, although the flies selected the highest temperatures in mornings, then moved to cooler temperatures later in the day, which the authors interpreted as a capacity to modulate fever reactions. The flies also responded to intensity of infection because higher-dose infections led to higher fever responses [13,14]. We suggest the work on *M. domestica* indicates rather small insects can express biologically meaningful, nuanced behavioral fevers, although Ballabeni et al. [15] reported that three *Drosophila* species did not fever in response to infection of the parasitic nematode, *Howardula aoronymphium*.

Insect immunity is a physiological trait that entails biological costs. Schmid-Hempel [16] recognized different kinds of costs, including evolutionary costs, maintenance costs, i.e., maintaining a selected level of readiness, and the costs of deploying immune systems in response to invasions. Several authors have reported on estimating the actual costs of immune reactions to invasion. Kraaijeveld and Godfray [17] selected *Drosophila melanogaster* for increased resistance to a parasitoid wasp, *Asobana tabida*. The increased resistance was accompanied by a sharp reduction in ability to compete for limited food resources. The authors considered the reduced competitiveness as a measure of the costs of increased resistance. In another system, leaf-cutting ants, *Acromyrmex octospinosus*, express paired exocrine glands that produce and secrete antibiotic compounds [18]. The authors blocked the glands of experimental ants with nail polish, then determined respiratory rates of experimental and control ants. They recorded significantly reduced metabolic rates in ants with a closed metapleural gland, which they interpreted as a measurable cost of immunity. Rolff and Siva-Jothy [19] expressed an evolutionary ecology view on insect immunity, which includes the possibility of trade-offs between traits. In this view, traits such as reproduction or migration may be traded off to support the immediate needs of immune reactions to infection.

Aside from trade-offs between costly physiological traits, we considered the possibility of physiological trade-offs within insect immune responses. The cellular immune mechanism, nodulation, can entail losses of millions of hemocytes in larger insects, which are replaced in hematopoietic organs [4]. Compared to the biological costs of cellular immunity, fever entails relatively low cellular losses or energy investment beyond sensing and shifting thermo-preferenda toward warmer temperature zones or assuming appropriate basking postures. Because behavioral fever may be a less costly alternative to the investment and replacement of large proportions of total hemocytes, we posed the hypothesis that some insect species trade off a quantifiable cellular immune reaction, nodulation, for behavioral fever reactions to infections. We tested our hypothesis using larvae of the fall armyworm, *Spodoptera frugiperda* and, in this paper, report on the outcomes of experiments designed to test our hypothesis.

## 2. Materials and Methods

### 2.1. Organisms

*S. frugiperda* larvae and their artificial diet were purchased from Benzon Research, Inc. (Carlisle, PA, USA). We received insect eggs in sealed diet trays in which the eggs hatched and the larvae developed. Individual larvae were reared on the provided medium in a growth chamber at 28 °C, 70% RH, under a 16L:8D photoperiod, our standard conditions. Fifth-instar larvae were used for all experiments. Bacteria, *Serratia marcescens* (SM), taken from a culture maintained at BCIRL, were seeded into LB medium and grown overnight to 10^9^ cells/mL in an orbital shaker at 31 °C. After dilution to concentrations indicated in Results, selected bacterial doses were injected into 5th instar larvae as described just below. 

### 2.2. Thermal Gradient Apparatus

In consultation with Dr. Zhang, BCIRL maintenance mechanic, Gordon Harper, designed and constructed two similar thermal gradient troughs (TGs), each 1 mL × 15 cm H × 15 cm W. The troughs were equipped with Perspex sliding lids that allowed observations without larval escape. The trough bottoms were covered with clean, dry sand and the troughs were wrapped in heating pads (25 cm on each end) to create a stable thermal gradient from 51 ± 1 °C at the ends to 27 ± 1 °C near the mid-point. They were operated at laboratory room temperature, approximately 25 °C. The heating pads were switched on for 2 h before each experiment to establish the same stable thermal gradients.

### 2.3. General Injection Protocol and Body Temperature Recording

Experimental larvae were narcotized on ice before injection. Bacterial challenges and pharmaceutical treatments were injected in 1 μL volumes using Hamilton 701 syringes (Hamilton, Reno, NV, USA). The needles were inserted into the larval bodies through inter-segmental sclerites with the needle held parallel to the body axis to avoid penetrating the alimentary canals. After injection, larvae were transferred onto their artificial diet in the 28 °C incubator to recover for 30 min, then moved into the TGs (from 27 ± 1 °C–51 ± 1 °C) and allowed to find their preferred temperature for times indicated in the Results section. Body temperatures of control and treated larvae were recorded using a Raytek MT2 infrared thermometer that was regularly calibrated against a mercury thermometer (Raytek Corp., Santa Cruz, CA, USA). Larval temperatures were recorded by placing the thermometer close to the thorax without contact. The TGs were cleaned with 70% ethanol after each experiment.

### 2.4. Bacterial Infections Lead to Behavioral Fever

In this experiment, we inoculated separate experimental groups with 10^6^ and 10^7^ bacterial cells/mL in the standard injection volume (1 μL). Control larvae were injected with the same volume of LB. After injection, the larvae were transferred to the 28 °C incubator to recover for 0.5 h post injection (PI), then transferred into a TG to observe their thermal preferenda and record the larval thoracic temperatures during 1, 2, 3, 4 and 5 h after injection.

### 2.5. PGs Mediate Behavior Fever

A little background will help with this experiment [2]. Prostaglandins (PGs) are oxygenated metabolites of three C20 polyunsaturated fatty acids, C20:3n-6, C20:4n-6 (arachidonic acid; AA) and C20:5n-3 The importance of PGs in human biology is indicated by the awarding of the 1982 Nobel Prize in Medicine or Physiology to three pioneers in the field. Among their many actions in humans, PGs mediate some headaches, which are relieved by pharmaceutical inhibitors of PG biosynthesis. Indomethacin (INDO) was discovered in 1963 and approved for use in humans in 1965. It relieves headaches by blocking PG biosynthesis. Some years later, scientists began researching PG actions in invertebrates and we now know PGs exert a wide range of actions, such as regulating Malpighian tubule physiology in adult mosquitos and many roles in insect immunity [1]. 

In the first step of PG biosynthesis, AA is released from cell membrane phospholipids by phospholipase A_2_. The availability of AA stimulates PG biosynthesis. In this experiment, we reduced PG concentrations by inhibiting their biosynthesis with INDO and in increased PG concentrations with AA.

We prepared stock solutions (SS) for this experiment, all in 500 μL solutions, and injected at 1 μL/larva. SS1, bacterial challenge treatment = 10^7^ SM cells/mL. SS2, PG biosynthesis inhibitor cocktail = 440 μL 10^7^ SM cells/mL + 50μL 1.8 mM INDO, + 10 μL EtOH. SS3, AA rescue cocktail = 440 μL 10^7^ SM cells/mL + 50 μL1.8 mM Indo + 10 μL 16.4 mM AA. SS4, negative control = 440 μL LB + 60 μL EtOH. Individuals in four groups of larvae were each treated with 1 μL of a stock solution, then returned to the 28 °C incubator. At 3, 23 and 47 h PI, larvae were placed into one of the TGs for two h per day over 3 days. Specific 2 h treatments were set at 3–5 h, 23–25 h, 47–49 h PI. At times PI noted in the Results section, larval temperature preferenda were recorded, n = 3 replicates in each control and experimental treatment, 15 larvae/replicate. We also recorded larval death, pupation, pupal death, adult emergence and pupal weight. Larvae that developed into deformed pupae and pupae that failed at adult emergence were discarded from the data set.

### 2.6. Duration of Behavioral Fever Influenced Development and Survivorship

Experimental larvae were injected with *S. marcescens* (10^7^ bacterial cells/μL in 1 μL/insect injections) and controls with the standard control treatment. All larvae were held in 28 °C incubators for 3 h, then transferred into a TG for 2, 4, 6 and 24 h, at which points we transferred the insects into a 28 °C incubator to record four parameters, larval death, pupation, pupal death and adult emergence each day. After pupation, we recorded the pupal weight on the first day, n = 3 biologically independent replicates, 15 insects/replicate. Larvae that developed into deformed pupae and pupae that failed at adult emergence were discarded from the data set. The percent emergence was calculated as % = emerged adults/total larval × 100.

### 2.7. Behavioral Fevers Led to Reduced Nodulation Reactions

We designed four treatments to compare the effect of behavior fever and imposed fever (heating) on nodulation reactions. Nodulation was assessed by anesthetizing treated larvae on ice for 30 min, then transferring them to a wax-bottom dissecting dish, abdominal surface up. The abdomen was opened by cutting the epidermis from head to tail and pinning the body open. The alimentary canal surface was examined for nodules, then removed. The remaining inner surface was gently washed with PBS to remove contaminants that could be mistaken for nodules, then the numbers and diameters of nodules were recorded and measured under a dissecting microscope (Leica, M165C, Buffalo Grove, IL, USA; Figure 1). Unlike other insect species, in which nodules are similar in size and can be directly counted [4], the nodules in fall armyworms are highly variable in size. To record nodules for this paper, we counted Indexed Nodules (IN) following the method of Zhang et al. [20]. In detail, we recorded the diameters of each nodule, which occurred in a size range of about 20 to over 200 µm. Each 30 µm was counted as one nodule (for example, 1–30 µm: one nodule; 31–60 µm: two nodules; 61–90 µm: three nodules…), such that a 1 to 30 µm nodule was counted as one nodule, a 40 µm nodule was indexed as 2 nodules and a 150 µm nodule was indexed as 5 nodules. Data are reported as numbers of IN/larva. The four treatments were as follows:(1)The influence of behavior fever for 5 h on nodulation reactions was determined. Control larvae were challenged with the standard SM treatment (10^7^ cells/mL in 1 μL/insect), then immediately transferred into the 28 °C incubator. Individuals in a second fever group were similarly challenged, then immediately transferred into the TGs. At 5 h PI, we collected the fevering larvae and counted numbers of IN/larva as described.(2)The influence of imposed fevers (heating) 5 h on nodulation reactions was determined. Control larvae were given the standard SM injection, then immediately transferred into the 28 °C incubator. The heating group of SM-treated larvae were immediately transferred into a 41 °C incubator for 5 h, and IN/larva were counted as described.(3)The influence of delayed behavior fever 4 h on nodulation reactions was determined. Control larvae received our standard SM treatment and were immediately transferred into the 28 °C incubator for 7 h. Experimental larvae were infected and incubated in the 28 °C incubator for 3 h immediately, then transferred into a TG. At 4 h after transfer, we counted IN/larva in fevering larvae.(4)The influence of delayed heating 4 h ahead on nodulation reactions was determined. Control larvae were injected with the standard SM treatment and immediately transferred into the 28 °C incubator for 7 h. Heating experimental larvae were infected and incubated in 28 °C for 3 h, then transferred into a 41 °C incubator for 4 h, then counted IN/larva in larvae with behavior fever.

### 2.8. Data Analysis

All data obtained are presented as mean ± SEM. Two groups of different treatments and controls were analyzed by Student’s-test, but the data among more than two groups were analyzed by one-way analysis of variance (ANOVA, version). Multiple treatments were compared using Tukey’s HSD test level. All the percentage data were square root arcsine transformed before ANOVA to meet the assumptions of normality. All statistical procedures were performed with SAS software (SAS, 1988).

## 3. Results

### 3.1. Bacterial Infections Lead to Behavioral Fever 

In TGs, control larvae selected approximately 30 °C temperature preferenda (Figure 2). Our standard bacterial challenge (10^7^ cells/mL) led to significant increases in body temperatures at 2 to 5 h PI, reaching a high of about 41 °C at 5 h PI. Smaller bacterial challenges (10^6^ cells/mL) also led to similar significant body temperatures increases, albeit with slower fever responses from 4 to 5 h. We recorded significant increases at 2, 3, 4 and 5 h PI (2 h: F = 4.27, df = 2, 42, *p* = 0.02; 3 h: F = 3.73, df = 2, 42, *p* = 0.03; 4 h: F = 8.01, df = 2, 42, *p* < 0.01; 5 h: F = 9.28, df = 2, 42, *p* < 0.01).

### 3.2. Four-Hour Fevers Led to Increased Survival 

We considered the influence of fevers on larval survival (Figure 3). Nearly all control larvae survived to adulthood, as expected. Less than 20% of bacteria-infected larvae survived, which significantly increased to about 50% in larvae allowed to fever for 4 h (*p* < 0.05), but not for 2, 6 or 24 h. Control treatments with LB, LB+INDO and LB+INDO+AA led to approximately 100% survival with no differences between larvae that did or did not fever.

### 3.3. Fever Duration Influenced Developmental Parameters 

S. marcescens infections led to increased larval, but not pupal developmental times and steeply reduced adult emergence (Table 1). Infections led to increased larval developmental periods, measured as times to pupation, from about 5.5 d in uninfected controls to 7.1 d in SM-treated control larvae. Four and 6 h fevers, but not other fever durations, led to significantly increased larval developmental time to pupation (8.3 and 8.6 days). Fevers did not influence pupal duration (7.4 to 7.9 days) although 4 h and 6 h fevers led to increased pupal weight (193–200 mg). Bacterial infection led to significantly reduced adult emergence, down from 100% in broth-injected controls to about 20%. Again, 4 h fevers led to significantly higher adult emergence at about 40%, although this is not true for insects that experienced fevers for 2, 6, and 24 h (20.0–22.2% emergence). Nonetheless, infections, per se, led to substantial reductions in adult emergence, down to about 18% in larvae that were not able to fever which was significantly lower than those of LB control and 4 h fever treatment (Table 1).

### 3.4. PG Signaling Mediates S. fruiperda Fevers 

As already seen, SM infection led to behavioral fever in experimental larvae (Figure 4). LB control treatments did not lead to fever reactions, while SM challenge led to substantial fevers which were sustained for the 49 h experimental time course. Treatments with bacteria plus INDO did not lead to fever reactions, from which we hypothesized that PGs mediate fevers. Adding the PG-biosynthesis precursor, AA, to the injection cocktails (SM+INDO+AA) led to restoration of the fever reactions. Control injections lacking SM (LB+INDO and LB+INDO+AA) did not lead to larval fevers. 

However, fevers in larvae treated with cocktails of SM+INDO and SM+INDO+AA did not lead to significant variance in survival, despite injecting INDO and AA (Figure 5). There were no significant differences among these six treatments (SM, SM+heating, SM+INDO, SM+INDO+heating, SM+INDO+AA, SM+INDO+AA+heating) (*p* > 0.05). In these LB treatments, injecting INDO and AA did not induce significant variance (*p* > 0.05). In a larger control experiment, we determined the influence of inhibiting, then restoring, PG biosynthesis on development. Table 2 reports that, with some exceptions, the indicated treatments did not influence pupal development times. We note that compared to the influence of SM infection with fevers, we recorded a small, albeit statistically significant increase in larval developmental times following treatments with a cocktail of SM+Indo+AA, without fever (from 4.5 to 6.7 days). Behavioral fever induced significant declines in larval duration after injection (*p* < 0.05), with SM+Indo+fever (5.0 days) vs. SM+Indo+no fever (6.0 days), SM+Indo+AA+fever (5.5 days) vs. SM+Indo+AA+no fever (6.7 days). Similarly, SM infections, but not the other treatments, led to reduced pupal weights, compared to the SM+fever treatment (from about 190 mg to 154 mg). As reported in Table 1, bacterial challenges led to reduced adult emergence and fever did not lead to recovery. No significant developmental differences occurred in the six LB controls reported in results with or without fever, Indo or Indo+AA (Table 3).

### 3.5. Fever Treatments Led to Substantial Reductions in Nodulation 

As seen in Figure 2 and Figure 4, larvae placed into a TG immediately after infection generated fevers by moving to higher temperature preferenda. Larvae that were not allowed to fever, by holding them at 28 °C, produced about 130 IN/larvae, while their counterparts that fevered produced less than 40 IN/larva (Figure 6A; F = 5.00, t = 4.71, df = 1, 16, *p* < 0.01). Similarly, imposing a high fever by transfer to a 41 °C incubator led to sharply reduced nodulation, down from about 75 IN/larvae to less than 20 (Figure 6B; F = 3.80, t = 4.19, df = 1, 19, *p* < 0.01).

We investigated the influence of fever reactions on nodulation in more detail by considering how lengthy delays (3 h) in fever generation influenced nodulation. We held infected larvae at 28 °C for 3 h, then transferred them to a TG or, in a parallel experiment, into a 41 °C incubator. In the TG choice experiment, nodulation was reduced in larvae that were allowed to generate behavioral fevers. This treatment led to smaller, yet significant reductions in nodulation, down by about 50% (Figure 6C; F = 0.049, t = 2.24, df = 18, *p* = 0.04). Nodulation was not reduced in larvae that were maintained at 28 °C, then transferred into incubators at 41 °C (Figure 6D; F = 0.29, t = −0.42, df = 1, 20, *p* = 0.68). 

## 4. Discussion

The data reported in this paper strongly support our hypothesis that fall armyworms, *S. frugiperda*, trade off a quantifiable cellular immune reaction, nodulation, for behavioral fever reactions to infections. Several points are germane. First, SM-infected larvae exhibited fever reactions to infection by selecting significantly warmer body temperatures after infections. Second, SM infections led to severely reduced larval survival, which was increased to about 50% in larvae that were allowed to fever. Third, fevers were inhibited in larvae treated with the PG biosynthesis inhibitor INDO. Fourth, fevers were restored in larvae that were treated with a cocktail of SM+INDO+AA, the precursor for PG biosynthesis. We note that control treatments lacking SM did not induce fevers. Fifth, the fevers were accompanied by substantially reduced nodulation reactions. Finally, fevers led to reduced rates of larvae development, increased pupal weights and survivorship to adulthood. Taken together, these points strongly support our hypothesis. 

Trade-offs are generally about allocations of limiting resources among traits. Our purpose is to focus on physiological, rather than evolutionary, trade-offs. In doing so, we recognize a substantial literature on evolutionary trade-offs. For example, Lochmiller and Deerenberg [21] concluded that competent immunological responses are necessary and costly to vertebrate hosts in terms of lifetime reproductive success. Specifically, they hypothesize that long-term survival would depend more on innate, rather than adaptive, immunity, particularly in short-lived species. Similarly, Harshman and Zera [22] clarify the cost of reproduction in terms of hormonal regulation and intermediary metabolism. They note that immunity and resistance to environmental stress are compromised by reproduction. With our focus on physiological trade-offs, we make the point that physiological trade-offs operate within individuals [23]. Trade-offs between immunity and reproduction are understood as resource allocation between two resource-demanding traits. Trade-offs among immune mechanisms are recorded as measurable changes in specific immune functions, although trade-offs can be more than simple one-to-one trades. Keehnen et al. [23] noted that immune trade-offs in *Pieris napi* larvae are involved in the suppression of non-immune processes, coupled with upregulating immune responses. We promote the idea that under appropriate environmental conditions, cellular immune mechanisms, in this case, nodulation, may be traded-off for behavioral fevers. 

Insects first evolved in the Early Ordovician period, about 479 million years ago (mya) and evolved flight in the Early Devonian, about 406 mya. [24]. They flourished over hundreds of million years despite virtually constant bacterial, fungal, protozoan and parasitoid infections and invasions, due in no small part to the evolution of complex, multifaceted immune systems. Among these systems, we focus this paper on two immune mechanisms, hemocytic nodulation and fever. While our data support our hypothesis, we recognize other immune mechanisms, such as the activation of prophenol oxidase, and elaboration of AMPs operating in the insect immunity. Future work will reveal how behavioral fever influences other aspects of insect immunity. 

Adamo [9] recognized that fever would be adaptive only in situations where two conditions are satisfied: one, the invader reduces reproductive fitness of its host, and two, the invader is sensitive to elevated host temperatures. Adamo tested this idea with adult crickets, *Acheta domesticus*. She learned that separate infections with the parasitoid fly, *Ormia ochrdacea*, with a gregarine gut protozoan, or the bacterium *S. marcescens* did not induce behavioral fevers. Infections with the intracellular parasite, *Rickettsiella grylli*, led to behavioral fevers, in which infected crickets in a temperature gradient shifted their thermal preference from about 26 °C to 32 °C. This may be adaptive for the crickets because the elevated host body temperatures led to decreased *R. grylli* survival. Similarly, house flies infected with fungus, *B. bassiana*, altered their fever behaviors during the day, shifting from the highest temperature areas in the morning to cooler areas over the day [14]. The point here is that insects do not express behavioral fever as a non-specific innate immune response to all infecting agents. To the contrary, fever may be a separate CNS-driven immune reaction to infection, one finely tuned to the temperature biology of hosts and their infectious agents. 

Our findings, that imposed 4 h fevers led to increased survival while the outcomes of shorter and longer imposed fevers did not, prompted questions. First, why did the shorter, 2 h fevers not extend survival? We suggest 2 h did not provide sufficient time to deplete enough of the infecting bacterial cells from the body to extend survival. And second, why did the longer fevers not extend survival? This may be a function of the behavioral ecology of insect fever. If, as just mentioned, fevers are closely tuned to host/invader biology, the *S. frugiperda* larvae in our study may not generate long, continuous fevers, but move in and out of warmer zones to achieve appropriate fever responses to a variety of infectious organisms and variable infection intensities. We also note that fevers are not uniformly beneficial to the fevering individuals. Most human fevers are due to sepsis, although there are other non-septic fevers that can be quite dangerous, even lethal, at the organismal level [7]. We speculate that insect behavioral fevers can also lead to dangerously high body temperatures. As seen in infected house flies, fevering insects can shift away from fever locations as appropriate through the day [13]. While there are no quantitative data on the experimental house fly body temperatures, they may shift to non-fevering locations to avoid a potential fever-driven pathology. We suggest more detailed research into insect behavioral fevers is necessary to grasp the more subtle aspects of their fevers. 

Our data confirm the idea put forth by Bundey et al. [25] that PGs mediate insect behavioral fever reactions to infection. This is one of several immune responses mediated by PGs [2]. In Bundey et al. [25], the temperature preferenda of untreated and control-treated fifth and sixth instar desert locusts was consistently set at approximately 30 °C, which increased following infection to about 32–38 *°C* by 1 h PI and held for the following 3 h. The rate of fever generation increased with increasing bacterial doses from 10^3^ to 10^4^ bacterial cells/dose. Based on work with tobacco hornworms, *Manduca sexta*, Howard et al. [26] reported that nodulation reactions, as numbers of nodules/larva, increased with increasing numbers of infecting bacteria. This has been recorded also for the greater waxmoth, *Galleria mellonella* [27], and the Madeira cockroach, *Leucophaea maderae* [28]. These findings support the idea that insects sense infection intensity and generate measured, proportional immune responses. And, again, indicate fevers are not non-specific innate immune reactions. Treating larvae with a cocktail of bacteria plus the PG biosynthesis inhibitor, INDO, blocked the infection-stimulated behavioral fevers, which were reversed in larvae treated with a rescue cocktail of bacteria+INDO+AA. Imposed heat treatments led to increased survivorship of infected larvae, albeit far below the survival of unchallenged insects. Bacterial challenge led to reduced larval, but not pupal, development rates. These points help identify PG signaling as one mechanism of behavioral fever. They also identify a physiological trade-off within immune mechanisms in *S. frugiperda* and, possibly other insect species that express behavioral fever.

The fever reactions led to substantial reductions in nodulation, down from about 130 IN/larva in larvae that did not fever to about 30 IN/larva in larvae that selected fever temperatures for 5 h. We infer that fever reactions are coupled to reduced nodulation. Our interpretation is that nodulation reactions are biologically expensive processes, involving immediate investment of very large numbers of hemocytes. For a quantitative example, bacterial challenge in tobacco hornworms, *Manduca sexta*, led to investments of millions of hemocytes in formation of over 100 nodules/larva following *S. marcescens* infections [4]. These investments are recorded as reduced numbers of circulating hemocytes, which in the short term are replaced by sessile hemocytes released from hemopoietic organs and later by the production of new hemocytes. Another indicator of the biological costs of hemocytes is seen in honey bees, *Apis mellifera*. Adult workers maintain substantial circulating hemocyte populations and effective hemocytic defenses which support nodulation and other hemocytic reactions to infection during the in-hive phases of their behavioral ontogeny. As they enter their high-energy foraging phase, they abandon virtually all hemocytes and hemocyte-based immunity, shifting to phenoloxidase-based immunity. We interpret these findings in terms of another trade-off among insect immune mechanisms [29]. There may be additional trade-offs within immune mechanisms, so long as they satisfy the conditions set out by Adamo [9]. We propose *S. frugiperda* larvae shift their immune strategy away from nodulation and toward behavioral fever under conditions in which the fever mechanism led to optimal fitness.

## 5. Conclusions

Our results support our hypothesis that, under appropriate conditions, at least some insect species respond to microbial infections with fevers and much reduced nodulation. The mechanisms of insect fevers differ from mammalian fevers in that insects generate behavioral fevers rather than increasing metabolic rates to fever levels. Insects fever by moving to warmer sites while reducing the costly investment of large numbers of hemocytes into cellular immunity. A range of rather subtle questions remain unexplored, such as how do insects decide whether to activate hemocytic immunity or fever? When insects are febrile, how do they select the appropriate fever temperature? How are the nodulation reactions reduced? As seen in the growth of insect biology generally, we foresee increased knowledge of insect fevers in the future. Insect immunology was once regarded as relatively simple. Possibly beginning with the discovery of the first AMP, cecropin, our appreciation of the many subtle features of insect immunity has grown. We note several advances. Some insect species express a certain immune memory. The genetic pathways that lead to the expression of AMPs are now known, the mechanisms of detecting infections have come to light, and immune signaling mechanisms, such as cytokine and prostaglandin signaling, are recognized. AMPs act not just in infections, but also in regulating the composition and abundance of microbiomes. The Antimicrobial Peptide Database (https://aps.unmc.edu (re-accessed on 7 November 2023)) lists 3569 AMPs of natural origin in six life kingdoms. Of these, about 10% (367) were discovered in insects. We conclude research into insect immunology is a fruitful, substantial field that will continually yield new insights into insect biology. 

## Figures and Tables

**Figure 1 insects-14-00864-f001:**
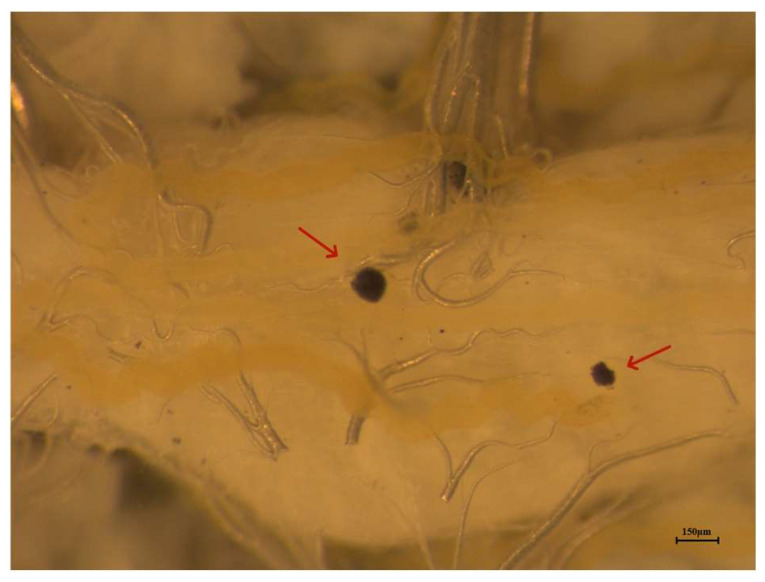
Melanized nodules in infected *S. frugiperda* larvae injected with bacteria, *S. marcescens* seen under a Leica dissecting microscope. Red arrows point to the nodules—the dark spots against the lighter internal organs and tissues. Microphotograph by Dr. Lei Zhang.

**Figure 2 insects-14-00864-f002:**
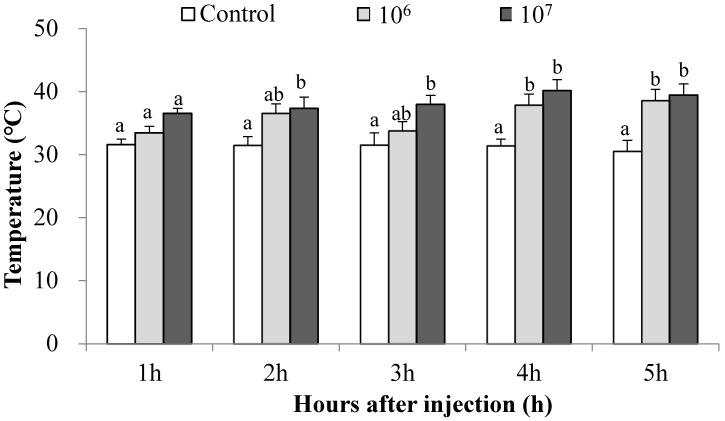
The preferred temperature of *S. frugiperda* during a 5 h period following injection with 10^6^ and 10^7^ cells/mL *S. marcescens* (SM). Mean temperature ± SEM, N = 15 for each treatment. Open bars, control insects injected with 1 μL LB; grey bars, 1 μL of 10^6^ cells/mL *S. marcescens*; black bars, 1 μL of 10^7^ cells/mL *S. marcescens*. Histogram bars annotated with the same letters (either a or b) indicate each three treatments on the same hours after injection (control, 10^6^ and 10^7^ cells/mL) are not statistically different at 5% level by Tukey’s HSD test (ab indicates no differences between these specific treatments).

**Figure 3 insects-14-00864-f003:**
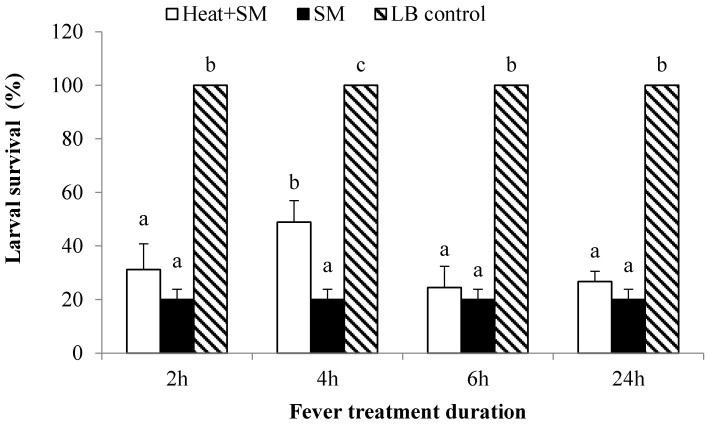
Effect of different duration of continuous behavior fever on mortality of *S. frugiperda* caused by injection *S. marcescens* (SM). Mean temperature ± SEM. Three replicate groups/treatment were generated, with each group containing 15 technical replications. Bars sharing the same letter are not significantly different within each three treatments (Heat+AM, SM and LB control) with the same fever treatment duration (2, 4, 6, and 12 h fever). SM and LB controls were cultured in 27 ± 1 °C incubator. Histogram bars annotated with the same letters (either a, b or c) indicate the treatments are not statistically different at 5% level by Tukey’s HSD test.

**Figure 4 insects-14-00864-f004:**
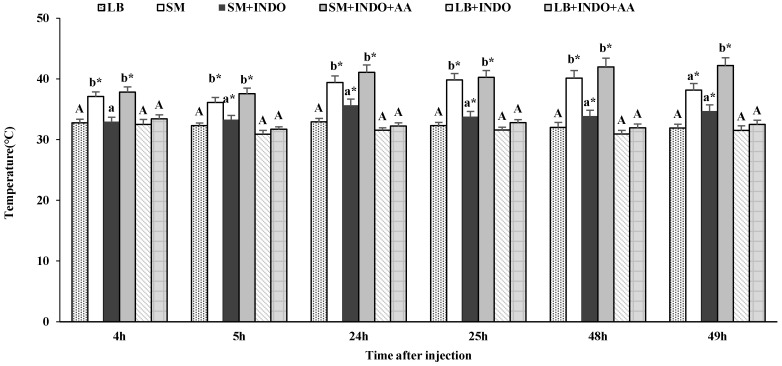
The influence of indomethacin (INDO) and arachidonic acid (AA) on the preferred temperature of *S. frugiperda* in discontinuous behavioral fever treatment. Data are presented as mean ± SEM. Bars sharing the same lower-case letter (a or b) are not significantly different among SM, SM+INDO and SM+INDO+AA treatments at 5% level by Tukey’s HSD test. Bars sharing the capital letter (A) are not significantly different among LB, LB+INDO and LB+INDO+AA control treatments at 5% level by Tukey’s HSD test. Bars sharing the asterisk are significantly different between SM and LB, SM+INDO and LB+INDO, SM+INDO+AA and LB+INDO+AA, respectively, at 5% level by Student’s *t*-test. Sample size from left to right is 45, 45, 45, 45, 45, 45, 45, 45, 45, 45, 45, 45, 30, 45, 34, 45, 28, 45, 30, 45, 34, 45, 28, 45, 26, 45, 30, 45, 22, 45, 26, 45, 30, 45, 22 and 45, respectively.

**Figure 5 insects-14-00864-f005:**
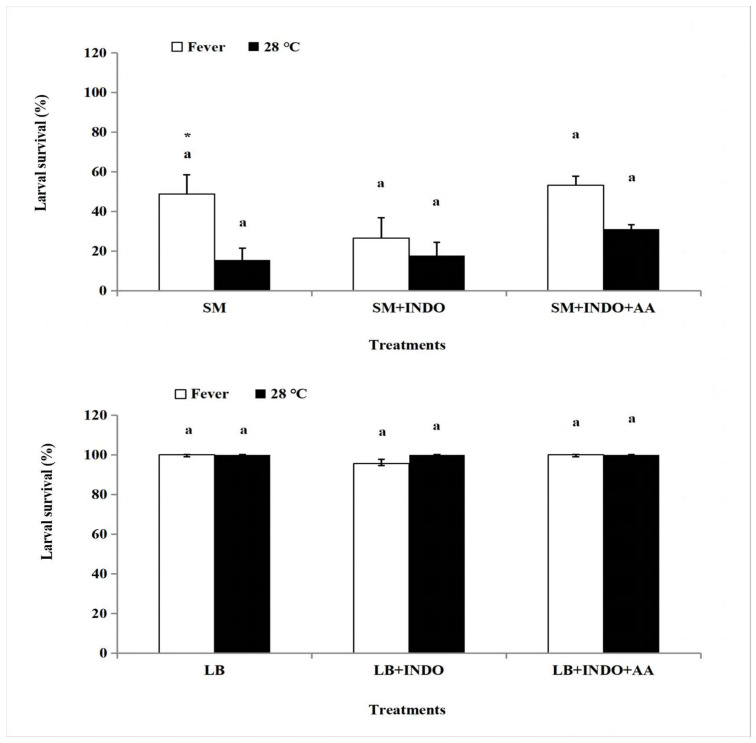
The influence of indomethacin (INDO) and arachidonic acid (AA) on the survivorship of *S. frugiperda* in discontinuous behavioral fever treatment. Data are presented as mean ± SEM. Replicate groups number is 3 and numbers of each replicate group is 15. Histogram bars sharing the same lower-case letter are not significantly different in three treatments (with the same treatment by fever or 28 °C) at 5% level by Tukey’s HSD test. Bars sharing the asterisk are significantly different between heating and 28 °C, at 5% level by Student’s *t*-test.

**Figure 6 insects-14-00864-f006:**
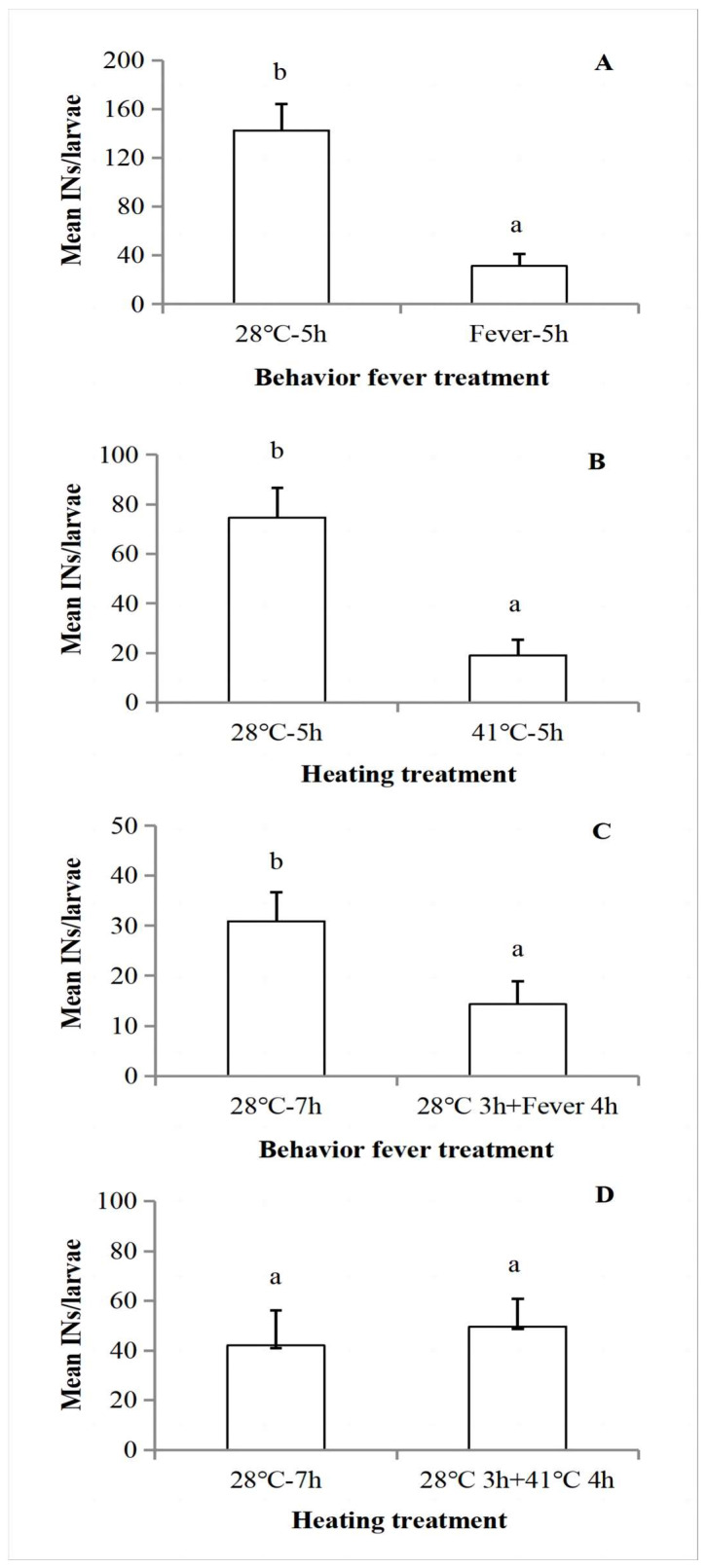
The influence of fever on nodulation. (**A**) The influence of immediate 5 h behavior fever choice for 5 h on nodulation in infected *S. frugiperda* by *S. marcescens*. Sample sizes for each treatment are all 9. (**B**) The influence of immediately imposed 5 h fever (heating) under 41 °C for on formulation of nodules in *S. frugiperda* after they were infected by *S. marcescens* for 3 h. Sample sizes for each treatment are 10, 11. (**C**) The influence of delayed (28 °C 3 h) behavior fever choice for 4 h on formulation of nodules in infected *S. frugiperda* after they were infected by *S. marcescens* for 3 h. Sample sizes for each treatment are all 10. (**D**) The influence of delayed (28 °C 3 h) imposed fever (heating) under 41 °C for 4 h on formulation of nodules in *S. frugiperda* after they were infected by *S. marcescens* for 3 h. Sample sizes for each treatment are all 11. Data are presented as mean ± SEM. Bars sharing the different letter are significantly different at 5% level by Student’s *t*-test.

**Table 1 insects-14-00864-t001:** The influence of the indicated fever duration on *S. frugiperda* pupal development. Data are presented as mean ± SEM. PI = post infection. Data in the same row annotated with the same lower-case letter are not significantly different (Tukey’s HSD test, *p* < 0.05). Data annotated with an asterisk are significantly different from SM controls (Student’s *t*-test, *p* < 0.05). (n) = number of biologically independent replicates. SM controls were injected with *S. marcescens* at 10^7^ cells/mL. Sample sizes are indicated in parentheses.

	LB Controls	SM Controls	2 h Fever	4 h Fever	6 h Fever	24 h Fever
**Time PI to pupation (d)**	5.5 ± 0.1 a (45)	7.1 ± 0.4 b (9)	7.9 ± 0.4 bc (14)	8.3 ± 0.2 c * (22)	8.6 ± 0.2 c * (11)	7.8 ± 0.3 bc (12)
**Pupal duration (d)**	8.0 ± 0.1 a (45)	7.4 ± 0.5 a (8)	7.9 ± 0.1 a (9)	7.5 ± 0.2 a (19)	7.4 ± 0.4 a (9)	7.9 ± 0.4 a (10)
**Pupal weight (mg)**	194.7 ± 2.4 ab (45)	183.1 ± 8.3 ab (9)	176.9 ± 6.5 a (14)	200.2 ± 6.1 b (22)	193.5 ± 7.6 ab (11)	174.0 ± 6.7 a (12)
**Adult emergence (%)**	100 a (3)	17.8 ± 4.4 b (3)	20.0 ± 7.7 b (3)	42.2 ± 4.4 b * (3)	20.0 ± 6.7 b (3)	22.2 ± 4.4 b (3)

Note: Student’s *t*-test compares SM Control (10^7^ *S. marcescens* cells/mL in 27 ± 1 °C) with heating treatments. Data are presented as mean ± SEM. Data in the same row sharing the same lower-case letter (a, b, or c) are not significantly different among LB control, SM control, 2, 4, 6 and 24 h heating treatments 5% level by Tukey’s HSD test (*p* < 0.05) (ab or bc indicate no differences between these specific treatments). Data annotated with an asterisk are significantly different between SM control and 2, 4, 6 and 24 h heating treatment, respectively (Student’s *t*-test, *p* < 0.05). Sample size from left to right is 45, 45, 45, 45, 45, 45, 45, 45, 45, 45, 45, 45, 30, 45, 34, 45, 28, 45, 30, 45, 34, 45, 28, 45, 26, 45, 30, 45, 22, 45, 26, 45, 30, 45, 22 and 45, respectively.

**Table 2 insects-14-00864-t002:** The influence of the indicated treatments on *S. frugiperda* developmental parameters.

	SM+ Fever	SM Controls	SM+Indo+ Fever	SM+Indo+ no Fever	SM+Indo+ AA+Fever	SM+Indo+AA+ No Fever
**Larval duration (d)**	4.5 ± 0.3 a (22)	4.9 ± 0.7 a (7)	5.0 ± 0.3 a (12)	6.0 ± 0.3 ab (8) *	5.5 ± 0.3 ab (23)	6.7 ± 0.4 b (14) *
**Pupal duration (d)**	8.3 ± 0.2 a (21)	7.6 ± 0.5 a (7)	7.8 ± 0.3 a (12)	7.6 ± 0.2 a (8)	8.3 ± 0.2 a (19)	7.8 ± 0.3 a (14)
**Pupal weight (mg)**	190 ± 3.3 b (22) *	154.4 ± 5.8 a (7)	183.8 ± 6.1 b (12)	177.8 ± 4.5 ab (8)	188.0 ± 2.8 b (23)	191.9 ± 6.3 b (14)
**Adult emergence (%)**	46.7 ± 10.2 a (3)	15.6 ± 5.9 a (3)	26.7 ± 10.2 a (3)	17.8 ± 5.9 a (3)	42.2 ± 8.0 a (3)	31.1 ± 2.2 a (3)

Data are presented as mean ± SEM. Data in the same one line sharing the same lower-case letter (a or b) are not significantly different among treatments at 5% level by Tukey’s HSD test. Data sharing the asterisk are significantly different between SM+fever and SM, SM+INDO+fever and SM+INDO, SM+INDO+AA+fever and LB+INDO+AA, respectively, at 5% level by Student’s *t*-test.

**Table 3 insects-14-00864-t003:** The influence of indomethacin (INDO) and arachidonic acid (AA) on the development of pupae of fall armyworm in discontinuous fever treatment.

	LB+ Fever	LB Controls	LB+Indo+ Fever	LB+Indo+No Fever	LB+Indo+AA+ Fever	LB+Indo+AA+No Fever
**Larval duration (d)**	3.7 ± 0.1 a (45)	3.9 ± 0.1 a (45)	3.9 ± 0.1 a (43)	4.0 ± 0.1 a (45)	3.8 ± 0.1 a (45)	3.8 ± 0.1 a (45)
**Pupal duration (d)**	8.0 ± 0.1 a (45)	8.0 ± 0.1 a (45)	7.7 ± 0.1 a (43)	7.8 ± 0.2 a (45)	7.8 ± 0.1 a (45)	7.7 ± 0.1 a (45)
**Pupal weight (mg)**	204.5 ± 3.9 a (45)	208.6 ± 2.6 a (45)	196.1 ± 3.4 a (43)	202.9 ± 3.1 a (45)	197.6 ± 3.1 a (45)	205.3 ± 3.4 a (45)
**Adult emergence (%)**	100 ± 0.0 a (3)	100 ± 0.0 a (3)	100 ± 0.0 a (3)	100 ± 0.0 a (3)	100 ± 0.0 a (3)	100 ± 0.0 a (3)

Data are presented as mean ± SEM. Data in the same row sharing the same lower-case letter are not significantly different among treatments at 5% level by Tukey’s HSD test.

## Data Availability

The data presented in this study are available on request from the corresponding author.

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
