# Peer review of "Trade-Offs among Immune Mechanisms: Bacterial-Challenged Spodoptera frugiperda Larvae Reduce Nodulation Reactions during Behavioral Fever"

_insects, 2023, doi:10.3390/insects14110864_

Round 1

Reviewer 1 Report

Comments and Suggestions for Authors

Manuscript Zhang et al Trade-offs among immune mechanisms: bacterial-challenged Spodoptera frugiperda larvae reduce nodulation reactions during behavioral fever

In their manuscript, the authors studied the role of a behavioral immune response (behavioral fever) compared to a cellular immune response (nodulation) in response to bacterial infection in an agriculturally relevant insect model, the fall armyworm caterpillar.

They show, that infected larvae prefer a higher temperature when offered a temperature gradient, and displayed slightly but significantly increased survival after infection when kept for 4 hours under higher temperature. They claim to show that prostaglandins are involved in fever generation by injecting a PG synthesis inhibitor cocktail and PG precursor arachidonic acid in addition to bacteria. However, the authors do not discuss how adding the precursor could rescue the effect of PG synthesis inhibition. They report reduced nodulation responses in animals with raised body temperature (either chosen or imposed).

Whereas the manuscript is generally well written and the experiments and results are explained and discussed well for the fever part, the nodulation section is less convincing. First of all, the methods should be explained in detail (how is the nodule score exactly calculated? give a formula!). The results must be illustrated by example photomicrographs of infected, sham-injected, fevered, and non-fevered animals displaying nodulation, in order to allow the reader to assess the quality of the preparations and measurements.

Figure 5 Labeling is not consistent (Fever treatment in B, Heating treatment in D). Nodules Scores should read nodule score (it is a single score).

It should be discussed, whether nodulation itself is a temperature sensitive process itself, or whether higher temperature simply reduces bacterial load, thus reducing nodulation indirectly. A useful test would have been to simply plate homogenized larvae at the end of the experiment to determine remaining bacterial load after different treatments.

Author Response

Dear Dr. Reviewer,

On behalf of all authors, I am submitting a revised version of our manuscript entitled, “Trade-offs among immune mechanisms: bacterial-challenged Spodoptera frugiperda larvae reduce nodulation reactions during behavioral fever” by L. Zhang, et al. We are submitting a clean copy and a marked copy that shows the major changes in our revision process. These include a re-written Abstract, a new section on biological costs of immunity, explanatory lines on the logic of the experiment in section 2.5, clarification of our method of determining Indexed Nodules, and additional lines in Discussion. The marked copy also highlights smaller changes, all developed in response to the review comments.

We are also submitting a copy of the review reports with our replies to each comment.

We feel the quality of the manuscript has been elevated to an acceptable level. However, if you decide more improvements are necessary, we stand ready to follow your guidance.

Sincerely,

Reviewer 2 Report

Comments and Suggestions for Authors

The article is very interesting. It is written in a clear and understandable way. Minor mistakes: line 215 - correct 10^7. Figure 2 -LB control bar could be a different color than Heat+SM. The plots in Fig 5 are far too large.

Author Response

(The authors gave the same response as above.)

Reviewer 3 Report

Comments and Suggestions for Authors

My section comments are as follows. 

Abstract

·      Overall Abstract is weak and needs to be rewritten. 

·      Abstract is missing techniques used in this experiment. I would suggest to add techniques, results output from this experiment, and conclusion drawn. 

Introduction 

·      Overall English of Introduction is good, but it has long paragraph such as Line 82-114. 

Materials and Methods

·      Line 132, Results must be written into results and check entire manuscript to address such kinds of typo errors. 

·      Line 135-136, BCIRL technician G. Harper. I would suggest to give reference or elaborate this line

·      I would suggest to include composition of artificial diet. 

·       Line 158, please be uniform in the entire manuscript cells/mL or at some places it is …./ml or µl but should be µL or uniform format.

·      I could not understand the logics of writing Results instead of results within the text body of manuscript.

·      Line 174: What does mean by Pupal Weight? Increase or Decrease? 

·      Line 178: Seems some values are incomplete or missing 

Results 

·      Line 228-229: cell/mL not cell/ml. Please follow the uniform pattern.

·      Line 303. Please follow same pattern e.g INDO or Indo…….

·      Line 348. I would suggest to write Figures instead of Figs….

Discussions

·      Line 409: I would suggest writing researcher, or author instead of using word she 

·      Line 412-Line 415. Reference is missing. Please include relevant references. 

·      Line 445: “Based on their work”. Whose work?? Please write clearly. 

Conclusions 

·      Line 482.  Our results support the hypothesis that……………

·      Line 490. Nodulation not italic..

·      Line 498-500. It is not part of the study and I think to add “The Antimicrobial Peptide
Database (https://aps.unmc.edu) lists 3,569 AMPs of natural origin in six life kingdoms.
Of these, about 10% (367) were discovered in insects”.  This part should be in Introduction or in Discussion at the relevant place instead of in conclusion.

Comments on the Quality of English Language

The Quality of English Language is fine,but need minor editing.

Author Response

(The authors gave the same response as above.)

Reviewer 4 Report

Comments and Suggestions for Authors

The article is interesting and attempts to demonstrate a trade-off between behavioral fever and immune response. The authors propose that their results demonstrate a trade-off between behavioral fever and nodulation. However, the introduction, the hypotheses, and the predictions still need to be improved. In addition, the results should be interpreted as a whole, considering the survival results and the proposed immune modulation.

The introduction could better explain that trade-offs arise because attributes are costly. This is the case for the immune response (Schmid-Hempel, 2021. Oxford University Press.). The immune response can be compromised within the attributes of the response to pathogens (i.e. fever versus nodulation or nodulation versus phagocytosis), but the immune response can also be compromised with reproduction and development. For example, males who have activated their immune response may be smaller and less competitive than large males who have not. If the behavioral response reduces developmental time, this could affect mating opportunities in adulthood. All of these theories have already been published. The authors can use them so that the reader unfamiliar with life histories can understand the reasoning behind their research from the outset.

Once you clarify that the response against parasites and pathogens is costly and compromises are expected, it can be said that organisms use physiological and behavioral responses to kill their pathogens. Subsequently, each could be explained separately. In the current version of the introduction, both topics are mixed in the first two paragraphs.

The idea of a trade-off between behavioral fever and immune response is interesting. You could cite several studies from Shelley Adamo because she has studied this topic in detail. For example, Adamo, S. A., & Lovett, M. M. (2011). Journal of Experimental Biology, 214, 1997-2004; Ferguson, L. V., & Adamo, S. A. (2011). Journal of Experimental Biology, 226, jeb244911.

Another important point is that there could not be evolutionary trade-offs behind their results, but strategic immunomodulation (see also Shelley Adamo's papers about the topic).

I would like to know why you only measured nodulation if you mention immune response in general. Behavioral fever may favor some attributes and reduce others. However, given that you only measured one, it is difficult to accept the conclusions you have reached.

Your hypothesis and predictions should be clearly explained.

Using live bacteria makes it impossible to discern the role of pathogens and hosts in the fever. Some pathogens manipulate hosts to move to higher or lower temperatures because that is where parasites thrive. Is this the case in your study?

Your introduction does not provide information about why you measured pupal duration, weight, adult emergence…

Specific comments:

Line 34 and 54. Reactive oxygen and nitrogen species are also part of the humoral immune response.

Lines 115-123. Indeed, may be costly because activates different parameters of the immune response and activates the metabolisms. In vertebrates, fever is costly.

Lines 119-123. Can you explain better your hypothesis and include your predictions? For example, one prediction is that fever increases resistance but decreases the developmental time or size.

Line 135-142. A figure may help to understand better your experimental design.

Please explain why you used AA, INDO… this will help to understand better the rationale of your experimental design. This can also be included in your predictions.

Why do you observe better survival at 4h and not before or after? 1 out of 4 treatments may contradict that fever favors resistance.

You can also discuss your results about development and size from a life history point of view, along with nodulation and survival according to treatments.

In general, the paper is interesting, but more work is needed. For example, can you add more immune response parameters to support a trade-off between immune response and fever firmly? For example, fever may reduce the cellular response but favor the humoral response. To support this, more immune markers should be recorded.

If you decide to keep only modulation, you should explain why you do not measure other immune response parameters.

Comments on the Quality of English Language

No comments.

Author Response

(The authors gave the same response as above.)

Round 2

Reviewer 1 Report

Comments and Suggestions for Authors

Revised Manuscript Zhang et al Trade-offs among immune mechanisms: bacterial-challenged Spodoptera frugiperda larvae reduce nodulation reactions during behavioral fever

Authors: Agree, we added details on counting nodules from Zhang et al., 2018.

Yes, some lines have been inserted in section 2.7 (by the way, why are there two sections 2.7?). These inserted lines contain some repetitions of what is written several lines before in the same paragraph and they contain some errors. In some places, nodules measure 30 µm, then they measure between 30 and 60 mm! The authors should carefully and thoroughly rewrite the paragraph rather than half-heartedly copying something into it. Furthermore, the process of measurement must be explained in detail. The authors do not provide photographs. How can someone measure the difference between 29 µm and 31 µm (one or two indexed nodules) without photographs and a computer? The authors write that they already have taken photographs 27 years ago in (Miller et al., 1996a) and Miller et al. (1996b). I don’t know these papers, and they are not referred to in the text and reference list. In any case, photos from old papers would not increase the credibility of today’s data. If the authors cannot provide a very good explanation and illustration of the precision of their measurements, and give examples from the treatment groups, I am afraid my concerns about the nodulation data are not alleviated, but exacerbated, and I cannot recommend publication.

Author Response

Dear Dr. Reviewer,

Dear Reviewer,

On behalf of all authors, I am submitting a second revision of our manuscript entitled, “Trade-offs among immune mechanisms: bacterial-challenged Spodoptera frugiperda larvae reduce nodulation reactions during behavioral fever” by L. Zhang, et al. Because relatively small changes were required, we are submitting a clean, rather than a marked, copy. Here are replies to the comments of each reviewer:

Sincerely,

Reviewer 4 Report

Comments and Suggestions for Authors

Thank you very much for your response. I see the improved version.

I have few comments:

1. It is not possible to discuss evolutionary costs without analyzing them in terms of fitness (reproduction and survival) and without taking into account different immune markers. If analyzed at a physiological level, it may be considered a cost but also an expenditure (Lochmiller, et al. 2000. Oikos, 88, 87-98; Kotiaho 2001. Biological Reviews, 76, 365-376; Harshman & Zera 2007. Trends in Ecology & Evolution, 22, 80-86.). It is also not possible to differentiate between commitment and immunomodulation (references that I suggested).

To prevent confusion, I recommend that you explicitly acknowledge that trade-offs are not considered in a strict evolutionary sense and consider providing mechanisms for these physiological trade-offs. 

2. You should reduce self-citation. For instance, instead of referencing from 1 to 5 in line 59, you can choose the one most significant reference. The same comment for line 545.

Author Response

Dear Reviewer,

On behalf of all authors, I am submitting a second revision of our manuscript entitled, “Trade-offs among immune mechanisms: bacterial-challenged Spodoptera frugiperda larvae reduce nodulation reactions during behavioral fever” by L. Zhang, et al. Because relatively small changes were required, we are submitting a clean, rather than a marked, copy. Here are replies to the comments of each reviewer:

Sincerely,
